# Space–Time Clustering and Socioeconomic Factors Associated with Mortality from Diarrhea in Alagoas, Northeastern Brazil: A 20-Year Population-Based Study

**DOI:** 10.3390/tropicalmed7100312

**Published:** 2022-10-18

**Authors:** Deanna dos Santos Lima, Wandklebson Silva da Paz, Álvaro Francisco Lopes de Sousa, Denise de Andrade, Beatriz Juliana Conacci, Flávia Silva Damasceno, Márcio Bezerra-Santos

**Affiliations:** 1Parasitic Diseases and Environment Graduate Program, Universidade Estadual de Alagoas—Campus II, Santana do Ipanema 57500-000, Alagoas, Brazil; 2Tropical Medicine Graduate Program, Universidade Federal de Pernambuco, Recife 50670-901, Pernambuco, Brazil; 3Global Health and Tropical Medicine (GHTM), Instituto de Higiene e Medicina Tropical, Universidade Nova de Lisboa, 1349-008 Lisbon, Portugal; 4Fundamental Nursing Program, Department of General and Specialist Nursing, University of São Paulo at Ribeirão Preto College of Nursing, Ribeirão Preto 14040-902, São Paulo, Brazil; 5Laboratory of Biochemistry of Tryps-LaBTryps, Department of Parasitology, Instituto de Ciências Biomédicas, Universidade de São Paulo, São Paulo 05508-040, São Paulo, Brazil; 6Medical Science Center, Universidade Federal de Alagoas, Arapiraca 57309-005, Alagoas, Brazil; 7Laboratory of Immunology and Molecular Biology, University Hospital, Universidade Federal de Sergipe, Aracaju 49060-108, Sergipe, Brazil

**Keywords:** public health, spatial analysis, diarrhea, gastroenteritis, Brazil

## Abstract

Acute diarrhea is the second leading cause of death among children in developing countries and is strongly related with the socioeconomic conditions of the population. In Brazil, data show a drop in the diarrhea mortality rate. Nevertheless, the northeastern region still has the most deaths. Considering this, we analyze high-risk areas for diarrhea- and gastroenteritis-related deaths, and their association with social determinants of health (SDH) in the state with one of the worst human development indicators in Brazil (Alagoas) between 2000 and 2019. We applied temporal, spatial, and space–time risk modelling. We used a log-linear regression model to assess temporal trends and the local empirical Bayesian estimator, the global and local Moran indices for spatial analysis. Spearman’s correlation was used to correlate mortality rates with SDH. A total of 3472 diarrhea-related deaths were reported during this period in Alagoas. We observed a decreasing time trend of deaths in the state (9.41/100,000 in 2000 to 2.21 in 2019; APC = −6.7; *p*-value < 0.001), especially in children under one year of age. However, there was stability among adults and the elderly. We identified two high-risk spatiotemporal clusters of mortality in inland municipalities. Lastly, mortality rates correlated significantly with 90% of SDH. Taken together, these findings indicate that diarrhea diseases remain a serious public health concern in Alagoas, mainly in the poorest and inland municipalities. Thereby, it is urgently necessary to invest in measures to control and prevent cases, and improve the living conditions of the poorest populations and those with the highest social vulnerability index.

## 1. Introduction

Acute diarrhea is a clinical manifestation characterized by the abnormal evacuation of liquid stools, usually resulting from an infection in the gastrointestinal tract and caused by a variety of microorganisms, such as bacteria and viruses. It is a frequent symptom of infectious intestinal diseases and is strongly associated with hygiene habits, and the ingestion of contaminated food and water, especially in poor populations, and those with limited access to safe drinking water and a sewage system [1,2].

There are globally nearly 1.7 billion cases of childhood diarrheal disease every year. More importantly, diarrheal diseases are responsible for about 525,000 deaths among children annually, and they are the second leading cause of death in those under five years old in developing countries. Clinically, acute diarrhea can last for several days and cause a severe loss of water, nutrients, and minerals. As a result, dehydration and the loss of electrolytes (sodium, chloride, and potassium) can lead to kidney failure and other serious medical problems, such as low blood pressure and hypovolemic shock. These clinical conditions can lead to death not only in children, but also the elderly, debilitated people, and those affected by comorbidities (e.g., HIV/AIDS) [1,3]. 

Regardless of the risk of diarrhea occurring in any community, cases and deaths are mostly reported in low-income countries, mainly from Africa, Asia, and Latin America. Social determinants of health (SDH), such as the human development index (HDI) and social vulnerability index (SVI), are associated with acute diarrhea and deaths [2,4]. Morbimortality from diarrhea, especially among newborns and children, is related to a lack of hygiene, and precarious conditions of basic sanitation and home infrastructure [5,6]. Additionally, in the poorest areas, there is a higher prevalence of bacterial (such as *Escherichia coli*) and viral (such as rotavirus, the most common cause of childhood diarrhea in industrialized and developing countries) [7] infections, and those caused by intestinal parasites (such as *Cryptosporidium* sp., *Giardia lamblia*, and *Entamoeba histolytic/dispar*) [2,8,9]. Considering this, a significant proportion of diarrheal disease can be avoided through safe drinking water, and adequate sanitation and hygiene [1].

In Brazil, data from the Ministry of Health have shown a drop in the mortality rate from diarrhea and infectious gastroenteritis in the last few decades [10]. Despite this reduction, the northeastern region still has the most deaths from acute diarrhea [11]. Furthermore, data show that infant diarrhea mortality rates in Brazil are higher in children under 1 year of age and poorer communities [12]. Remarkably, the state of Alagoas (Northeastern Brazil) has one of the worst socioeconomic indicators in the country. The state has serious deficiencies in the sewage system, water supply, and literacy rate, and the HDI is the lowest in Brazil (0.683) [13].

Considering this, we analyze high-risk areas of diarrhea- and gastroenteritis-related deaths, and their correlation with SDH in the state of Alagoas, between 2000 and 2019, using space–time risk modeling. This approach allows for monitoring the impact of diarrheal diseases in space and time [14,15,16,17]. As a result, this could significantly contribute to health systems in planning disease control strategies, and reduce the mortality rate from diarrhea and gastroenteritis, mainly among children. 

## 2. Materials and Methods

### 2.1. Design and Study Area

We carried out an ecological and population-based study using spatial, temporal, and spatial–temporal analysis techniques. The units of analysis were the 102 municipalities in the state of Alagoas, and the study period was from 2000 to 2019. We assessed all deaths caused by diarrhea or infectious gastroenteritis registered in Code A09 of the 10th Revision of the International Statistical Classification of Diseases and Related Health Problems (ICD-10). 

The state of Alagoas is one of the 27 federative units in Brazil and is located in the east of the northeast region. The state can be geographically divided into the metropolitan area and inland municipalities (Figure 1). The metropolitan area includes the capital, Maceió, and 12 other municipalities. The interior of the state encompasses all the 89 other municipalities, and they are the ones with the lowest HDI and highest SVI. In addition, the state is the second smallest federative unit in Brazil and has an estimated population of 3,351,343 inhabitants [13,18].

### 2.2. Data Source and Collection 

Data regarding deaths from diarrhea and gastroenteritis according to the municipality of residence were collected from the Mortality Information System (SIM) of the Brazilian Ministry of Health. SIM data are in the public domain and can be obtained from the website of the informatics department of the Unified Health System (DATASUS) [11]. Data on the population estimates of the municipalities between 2000 and 2019 were obtained from the Brazilian Institute of Geography and Statistics (IBGE), considering information from the national population census in 2000 and 2010 and official estimates for the intercensus years [18]. Information on the socioeconomic indicators was extracted from the United Nations Development Program (UNDP) [19], and from the Institute for Applied Economic Research (IPEA) [20] according to the 2010 Brazilian demographic census indicators. Lastly, the digital cartographic mesh of the state of Alagoas in shapefile format was obtained from the geographic projection system in latitude/longitude (Geodesic Reference System, SIRGAS 2000) available on the IBGE website. 

### 2.3. Variables and Indicators

The assessed variables in this study were:Number of deaths from diarrhea and gastroenteritis registered in the 102 municipalities of Alagoas, and categorized according to the sociodemographic variables of sex, age group, ethnicity or color, and educational level.Annual gross mortality rates by municipality, inland municipality, and metropolitan area per 100,000 inhabitants (we considered the number of annual deaths as the numerator and the corresponding population as the denominator).Average mortality rates by municipality and in the state of Alagoas per 100,000 inhabitants (we considered the number of total deaths as the numerator and the central population of the period as the denominator, divided by the number of selected years).Proportion of municipalities with diarrhea- and gastroenteritis-related deaths.

### 2.4. Temporal Trend Analysis

Temporal trend analysis was performed using the segmented log-linear regression model through the Joinpoint Regression Program version 4.7.0.0. This method allows for verifying changes in the indicator trend over time by adjusting data from the fewest possible joinpoints (inflection points) and tests whether the inclusion of more joinpoints is statistically significant. Thus, the time series can present an increasing, decreasing, or stable trend, and even different trends in sequential sections. Annual mortality rates from diarrhea and gastroenteritis were considered to be dependent, and the years of study were the independent variables. First, we performed the Monte Carlo permutation test to select the best model for inflection points, applying 9999 permutations and considering the highest coefficient of the determination of residuals (R2). Thereafter, to describe and quantify time trends, we calculated the annual percentage changes (APC) and their respective confidence intervals (95% CI). Once more than one inflection points had been detected during the study period, the average annual percentage changes were calculated (AAPC). The time series can present an increasing trend (positive and statistically significant APC or AAPC), decreasing trend (negative and statistically significant APC or AAPC), and stable trend (nonsignificant trends regardless of APC or AAPC values) [21]. Temporal trends were considered to be statistically significant when APC or AAPC presented a *p*-value < 0.05, and its 95% CI did not include the zero value. 

### 2.5. Spatial Analysis of Mortality Rates from Diarrhea and Gastroenteritis in Alagoas

First, we constructed spatial distribution maps of the crude and smoothed mortality rates from diarrhea and gastroenteritis in Alagoas. To smooth out the crude rates, we used the local empirical Bayesian estimator. This tool minimizes the instability caused by the random fluctuation of cases in space [22]. Crude and smoothed mortality rates were represented on choropleth maps and stratified into five categories of equal intervals: (a) 0–3.0, (b) 3.0–6.0, (c) 6.0–9.0, (d) 9.0–12, and (e) ≥12 per 100,000 inhabitants. Subsequently, to verify whether the spatial distribution of deaths occurred in a space-dependent manner, we performed the spatial autocorrelation analysis of crude mortality rates by calculating the univariate Moran global index (I). In this step, we elaborated a spatial proximity matrix obtained with the contiguity criterion, with a significance level of 5%. This index ranges from −1 to 1: values between 0 and 1 indicate positive spatial autocorrelation, between −1 and 0 indicate negative spatial autocorrelation, and values that cross zero indicate spatial randomness [23].

Lastly, once the positive spatial autocorrelation of the data had been identified, we evaluated the occurrence of local autocorrelation by calculating the local Moran index (or local indicators of spatial association—LISA). This method aims to identify the existence of spatial dependence and risk patterns in four parameters: Q1 (high/high) and Q2 (low/low), which indicate municipalities with spatial association and similar values among their neighbors; and Q3 (high/low) and Q4 (low/high), which indicate municipalities with discordant values between neighbors and without spatial association [24]. We carried out spatial statistical analyses using TerraView software version 4.2.2 (INPE. São José dos Campos, Brazil). Maps were produced using QGIS software version 3.12 (Open Source Geospatial Foundation, Beaverton, USA). Results were considered to be statistically significant if a *p*-value < 0.05 was obtained.

### 2.6. Space–Time Scanning Analysis and Identification of Risk Clusters

We applied a space–time retrospective and prospective scans to identify the existence of clusters of deaths from diarrhea and gastroenteritis in the municipalities of Alagoas during the study period. For that, we used the Poisson probability distribution model [15,23] while considering the following parameters: aggregation time of 1 year, minimum of 5 deaths, no overlapping clusters, circular clusters, maximal spatial cluster size of 50% of the population at risk, and maximal temporal cluster size of 50% of the study period [25]. 

The primary and secondary clusters were detected using the log-likelihood ratio test (LLR), and represented in choropleth maps and tables. Additionally, we calculated the relative risks (RR) of mortality from diarrhea and gastroenteritis, considering each municipality and clusters related to its neighbors. Data that presented *p*-value < 0.05 using 999 Monte Carlo simulations were considered to be statistically significant. Retrospective and prospective scan analyses were performed using SatScan software version 9.6 (National Cancer Institute, New York City, NY, USA), and maps were constructed using QGIS software version 3.12.

### 2.7. Correlation Analysis between Deaths from Diarrhea and Gastroenteritis and SDH

Initially, we selected 20 socioeconomic indicators from the Atlas of Human Development and Social Vulnerability in Brazil (UNDP and IPEA) [19,20]. Thereafter, we applied the D’Agostino and Pearson omnibus normality test to assess the parametric distribution of the data. Considering that the results were nonparametric, we performed the Spearman correlation test (Rho) [26]. The average crude mortality rate for the entire period of each municipality was used as the dependent variable and correlated with the selected socioeconomic indicators (independent variables). Statistical analyses were performed using GraphPad Prism software version 8.0 (GraphPad Software, San Diego, CA, USA, www.graphpad.com, accessed on 10 October 2022). Data were considered to be statistically significant if a *p*-value < 0.05 was obtained.

### 2.8. Ethical Considerations

We used secondary and public domain data. There is no possibility of identifying the study subjects; therefore, the informed consent was dispensed and it was not necessary to submit the project for consideration by a research ethics committee. However, national and international ethical recommendations, and the rules of the Declaration of Helsinki were followed.

## 3. Results

Our analyses indicate that a total of 3472 diarrhea- and infectious-gastroenteritis-related deaths occurred in Alagoas between 2000 and 2019. Regarding the epidemiological characteristics of these deaths (Table 1), we observed a higher percentage among men (53.37%), those aged < 1 year old (*n* = 1853; 46.86%), mixed-race individuals (*n* = 1707; 49.16%), and the illiterate (*n* = 743; 21.4%). Surprisingly, 34.79% (*n* = 1208) of these deaths occurred in the elderly (≥60 years).

The mortality rate in Alagoas showed a decreasing time trend (9.41/100,000 in 2000 to 2.21 in 2019; APC = −6.7%; *p*-value < 0.001; Table 2). Likewise, we observed a decreasing trend in both sexes (male and female, APC = −7.6% and −5.1%, respectively; *p*-value < 0.001). Concerning the age groups, those <1 year and 1 to 4 years old showed a decreasing trend (APC = −13.5 and −10.1, respectively; *p*-value < 0.001). Impressively, the mortality rate in <1-year-olds dropped from 275.82/100,000 in 2000 to 14.99 in 2019. On the other hand, the other age groups showed stability in deaths from diarrhea and gastroenteritis. Regardless of the stable trend, the mortality rate in those aged from 20 to 39 years increased from 0.92 in 2000 to 8.51 in 2019. Furthermore, we observed a decreasing trend regardless of the state area (metropolitan area or inland municipalities, APC = −4.0 and −3.3, respectively; *p*-value < 0.001). There was also a drop in the proportion of municipalities with death records in the state (APC = −3.3; *p*-value < 0.001). 

Subsequently, to identify areas of a higher occurrence of deaths from diarrhea and infectious gastroenteritis, we assessed their spatial distribution in the municipalities of Alagoas (Figure 2A–C). We observed records of deaths in all municipalities in the state. However, 11 municipalities had a mortality rate from 9 to 12, and 3 of >12/100,000 inhabitants. Interestingly, the municipalities with the highest mortality rates were from the interior of the state (Figure 2A). When we applied spatial-data-smoothing techniques that also considered data from neighboring municipalities, there was a reduction in the number of municipalities with a death rate from 9 to 12 and >12/100,000 inhabitants (1 and 7, respectively; Figure 2B). 

Spatial autocorrelation analysis shows a positive and significant spatial dependence (I = 0.2995; *p*-value = 0.006) in the deaths among municipalities of the state. As expected, those that presented a Q1 dependency pattern (high/high, in red) were inland municipalities (Figure 2C). Alternatively, the 11 municipalities presented a Q2-type pattern (low/low, in dark green) were all located on the coastal strip, and 6 from the metropolitan area of the state.

**Figure 2 tropicalmed-07-00312-f002:**
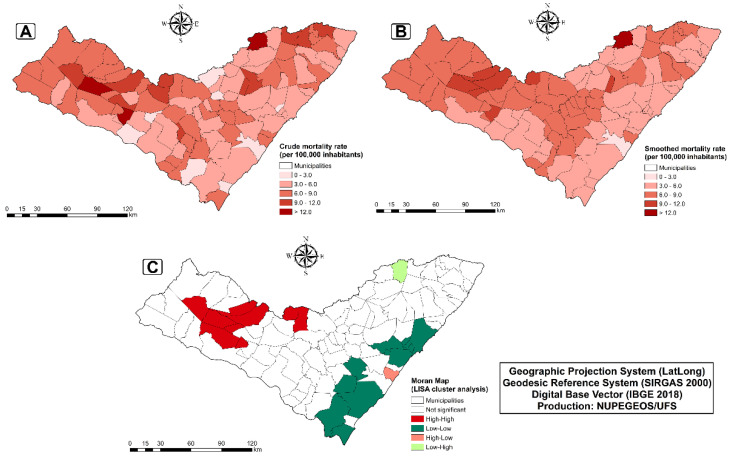
Spatial distribution of mortality rates due to diarrhea and infectious gastroenteritis in the state of Alagoas: Crude mortality rate (**A**); Smoothed mortality rate (**B**) and Moran map (**C**) (spatial autocorrelation). A). The primary cluster covered 67 municipalities considered to be at high risk from 2000 to 2008 (mortality rate = 10.11/100,000 inhabitants; RR = 3.67; Table 3). The secondary cluster included 20 municipalities between 2000 and 2007 (12.51/100,000; RR = 2.37). Importantly, all municipalities in those two clusters were from the interior of the state.

Similarly, we identified two spatiotemporal clusters through prospective spatiotemporal scan analysis (Figure 3B). Nonetheless, the primary cluster identified only one high-risk municipality (Santana do Ipanema) between 2013 and 2019 (9.62/100,000; RR = 1.74). Likewise, the secondary cluster included one municipality (Jacaré dos Homens) between 2015 and 2019 (22.53/100,000; RR = 4.06), evidencing that these two municipalities still had active conglomerates at the end of the study period (2019).

Lastly, we carried out correlation analysis on the mortality rate due to diarrhea and gastroenteritis with the SDH of the municipalities of Alagoas. We found that 18 (90%) indicators showed significant correlation with the mortality rate (Table 4). Importantly, we observed a positive correlation with the percentage of children aged 0–5 who did not attend school (Rho = 0.4012; *p*-value < 0.001), mortality up to 1 year of age (0.3449; *p*-value = 0.004), SVI (0.3374; *p*-value = 0.001), GINI index (0.2976; *p*-value = 0.008), and the percentage of people in households with inadequate water supply and sanitation (0.2975; *p*-value = 0.002). On the other hand, there was an inverse correlation with life expectancy at birth (–0.3470; *p*-value = 0.001), MHDI (–0.3455; *p*-value = 0.004), and the per capita income of those vulnerable to poverty (–0.3289; *p*-value = 0.005).

## 4. Discussion

Acute diarrhea is the second leading cause of death in children under the age of five and still represents a serious public health concern, mainly in low-income countries. Despite being preventable and treatable, more than half a million children under the age of five die annually from diarrheal diseases. Herein, our analyses show that the highest percentages of deaths caused by diarrhea and infectious gastroenteritis in Alagoas occurred among children under 1 year of age (46.86%) and in the elderly (34.79%). Interestingly, there was a significant reduction in deaths in Alagoas’ municipalities, especially among children under 1 year of age. On the other hand, there was an increase in the mortality rate among young adults, and stability in people aged over 60 years. Additionally, spatial analyses show high-risk clusters of mortality in inland municipalities and a correlation with most of the SDH assessed in this study. 

Several age-related factors could contribute to increasing susceptibility to, and the occurrence and severity of diarrheal diseases [27]. For example, high susceptibility among children, especially those under 5 years of age, may be associated with the immaturity of the immune system. Furthermore, these children are more exposed to pathogens due to a lack of personal hygiene [27]. Otherwise, the high number of deaths from diarrhea in the elderly is related to several factors, such as the senescence of the immune system, changes in the intestinal microbiota, a higher prevalence of comorbidities that affect the gastrointestinal tract, neoplasms, and the more frequent use of antibiotics and immunosuppressants [3,27]. 

Time series studies using spatial analysis techniques and an association with socioeconomic variables are required for a better understanding of the dynamics of diseases in populations in space and time [4,28,29,30,31]. Despite the high number of diarrhea-related deaths that occurred over two decades in Alagoas compared to other Brazilian states, we observed significant decreasing trends in most of the variables assessed in this study. Likewise, in a nationwide population-based study, the authors observed a decreasing time trend in the mortality rate and hospital admissions for diarrheal diseases in all Brazilian regions between 2000 and 2010. Importantly, the Northeast showed the greatest reduction in the percentage of deaths from diarrhea in the country [12,32]. 

Several factors may have contributed to reducing deaths from diarrhea, mainly among children in Alagoas: (i) significant reduction in the poverty rate in Brazil in the last two decades; (ii) income distribution programs such as Bolsa Família; (iii) improvements in the sewage system and drinking water supply; (iv) a decrease in child malnutrition; (v) expanding access to public health services; (vi) campaigns to encourage breastfeeding; and (vii) increased vaccination coverage against rotavirus [2,5,10,33,34]. 

Vaccination against rotavirus was implemented in Brazil in 2006 and is one of the most important factors associated with the decrease in child morbidity and mortality from diarrheal diseases [35]. Corroborating this, a study conducted by Santos and colleagues [7] demonstrated the impact of the rotavirus vaccine on diarrhea episodes among children from a municipality of the state of Sergipe, Northeast Brazil. In that study, the authors observed a decline in the diarrhea mortality rate, especially in the under-1-year age group. This result shows that vaccination against rotavirus contributes to reducing the mortality rate from diarrhea in the infant group [33].

Likewise, data show that the expansion of primary health care is one of the factors that has most contributed to the reduction in mortality from diarrheal diseases among children in Brazil [10,34]. Among the programs implemented by the Brazilian Unique Health System (SUS), the Family Health Strategy (ESF) provides the population with better access to public health services [36]. The ESF was implemented by the Brazilian Ministry of Health in 1994, and includes a series of actions to prevent infectious diseases and basic care for the population through multidisciplinary health teams composed of physicians, nurses, nursing technicians, and community health workers. As a result, data show that the ESF markedly reduced the mortality of children up to 1 year of age regardless of socioeconomic status. In 1994, the infant mortality rate in Brazil was 43 deaths per 1000 live births. In 2017, this index dropped to 12.4/1000. More importantly, in 2011, the country reached the Millennium Development Goal 4 target and reduced the infant mortality rate to 15.3/1000. 

Concerning the spatial analysis data, diarrhea mortality records were observed in all municipalities of Alagoas. Nevertheless, inland municipalities showed the highest death rates. Similarly, the spatiotemporal scan also identified high-risk clusters of deaths from diarrhea in inland municipalities. Regarding these findings, those municipalities usually have the worst socioeconomic indicators (sewage system and safe drinking water), and, along with less access to health services, this significantly increases the risk of gastrointestinal infection and acute diarrhea. Similar data were reported in other studies, with a higher occurrence of diarrhea-related deaths in inland municipalities in the states of Maranhão, Ceará, Piauí, and Pernambuco, and in northern and central–western region states [2,6,37]. 

Unquestionably, a significant proportion of diarrheal diseases can be prevented through SDH improvements such as safe drinking water, and adequate sanitation and hygiene1. We demonstrated a positive correlation of the mortality rate from diarrhea and gastroenteritis with the percentage of children aged 0–5 who do not attend school, mortality up to 1 year of age, SVI, GINI index, and percentage of people in households with inadequate water supply and sanitation. Conversely, there was an inverse correlation with life expectancy at birth, MHDI, and per capita income of vulnerable to poverty. Unfortunately, Alagoas is the state with the lowest MHDI in Brazil, and it still has low indicators regarding the sewage system, education, and access to safe drinking water [13,14]. Altogether, our findings corroborate the correlation between socioeconomic conditions and the occurrence of diarrheal deaths.

Our analyses show a significant reduction in the mortality rate in children from diarrheal diseases in the state of Alagoas between 2000 and 2019. Our study, however, has some limitations that should be mentioned. As an ecological study using secondary data, this may have resulted in the loss of some information due to under-reporting or misdiagnosis. Likewise, delays in the computerization of disease notification systems in many municipalities in the state can compromise the quantity and quality of presented information, especially in remote years. 

## 5. Conclusions

Taken together, our analyses show the temporal trend and map the spatial distribution of indicators of mortality from diarrhea and gastroenteritis in the state of Alagoas. There was a significant reduction in deaths from diarrhea among children under 1 year of age. Nevertheless, the increase in mortality rate among young adults, and stability in the elderly demonstrate that diarrheal diseases are still an important public health concern in Alagoas. Additionally, we identified a correlation between mortality rates and most SDHs in the state. Considering this, public policies aimed at combating social inequalities and improving the state’s socioeconomic indicators are essential to reduce cases and deaths from diarrheal diseases and other neglected tropical diseases, especially in the most vulnerable populations (such as children and the elderly). Regardless of the achieved progress, Brazil has faced political, social, and economic crises in recent years. Along with that, there have been expressive cuts in investments in education and health programs, and an increase in unemployment, starvation, and the population on the poverty line. More importantly, with the COVID-19 pandemic, the country’s socioeconomic situation was further aggravated. As a result, progress achieved in the last two decades may be lost, and the Brazilian population may once again suffer from hunger and the public health problems that affected it before the 2000s. 

## Figures and Tables

**Figure 1 tropicalmed-07-00312-f001:**
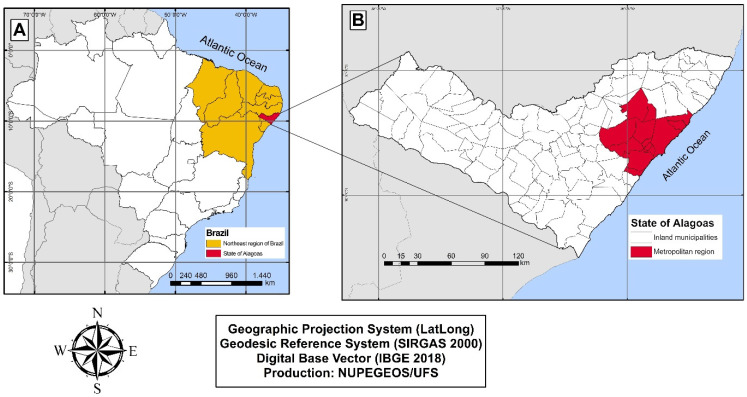
Map of the study area. (**A**) Map of Brazil highlighting the Northeast region (in yellow) and highlighting the state of Alagoas (in red). (**B**) Map of the state of Alagoas divided into metropolitan region (in red) and inland municipalities (in white).

**Figure 3 tropicalmed-07-00312-f003:**
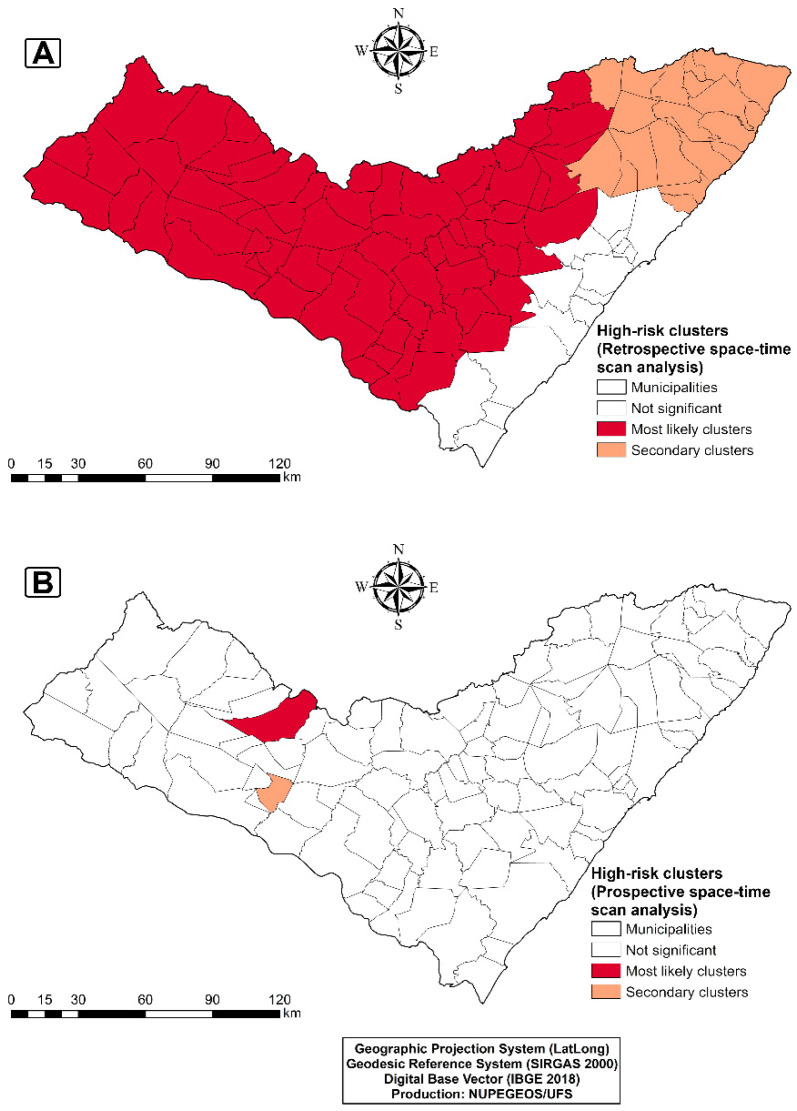
Spatiotemporal statistical analysis and clusters of deaths from diarrhea and infectious gastroenteritis in municipalities in the state of Alagoas, Northeastern Brazil, between 2000 and 2019. (**A**) Retrospective spatiotemporal scanning. (**B**) Prospective spatiotemporal scanning.

**Table 1 tropicalmed-07-00312-t001:** Epidemiological characteristics of deaths from diarrhea and infectious gastroenteritis in the state of Alagoas, Northeast Brazil between 2000 and 2019.

Variables	N	%
Sex		
Male	1853	53.37
Female	1619	46.63
Age group (year)		
<1	1627	46.86
1 to 4	251	7.23
5 to 19	55	1.58
20 to 39	89	2.56
40 to 59	242	6.97
≥60	1208	34.79
Race and ethnicity		
Caucasian	619	17.83
African descent	121	3.49
Asian descent	8	0.23
Mixed race	1707	49.16
Amerindians	15	0.43
Missing data	1002	28.86
Educational level (years)		
None	743	21.4
1 to 7	268	7.72
≥8	58	1.67
Missing data	2403	69.21

**Table 2 tropicalmed-07-00312-t002:** Temporal trend of mortality rates from diarrhea and infectious gastroenteritis according to sociodemographic variables in the state of Alagoas, Northeast Brazil between 2000 and 2019.

Indicators/Variables	Mortality Rate Per 100,000 Inhabitants		APC (95% CI)	*p*-Value	Trend
	2000	2019			
Alagoas	9.41	2.21	−6.7 (−8.6 to −4.7)	<0.001	Decreasing
Sex					
Male	10.04	1.93	−7.6 (−9.7 to −5.4)	<0.001	Decreasing
Female	8.47	2.47	−5.1(−7.2 to −3)	<0.001	Decreasing
Age group (years)					
<1	275.82	14.99	−13.5 (−15.3 to −11.6)	<0.001	Decreasing
1 to 4	12.23	0.49	−10.1 (−12.2 to −7.9)	<0.001	Decreasing
5 to 19	0.10	0.32	1.3 (−7.1 to 10.6)	0.787	Stable
20 to 39	0.92	8.51	−3.5 (−8.2 to 1.5)	0.194	Stable
40 to 59	3.44	1.76	−2.5 (−5.8 to 0.9)	0.187	Stable
≥60	14.22	14.74	11.9 (−2 to 27.9)	0.183	Stable
State area					
Metropolitan area	7.64	1.86	−7.3 (−9.4 to −5.3)	<0.001	Decreasing
Inland municipalities	10.10	2.37	−4.0 (−6.6 to −1.3)	<0.001	Decreasing
Proportion of municipalities with deaths	0.61	0.34	−3.3 (−4.6 to −1.9)	<0.001	Decreasing

**Table 3 tropicalmed-07-00312-t003:** Retrospective and prospective spatiotemporal scanning and risk clusters of the mortality rate from diarrhea and infectious gastroenteritis, in the state of Alagoas, Northeast Brazil between 2000 to 2019.

Clusters	Period	Municipalities	Deaths	Expected Deaths	Annual Mortality Rate *	RR	LLR	*p*-Value
Retrospective								
1	2000–2007	67	1336	733	10.11	3.34	270.85	<0.001
2	2000–2008	20	280	123.77	12.53	2.37	76.05	<0.001
Prospective								
1	2013–2019	1	32	18.5	9.61	1.74	4.06	<0.021
2	2015–2019	1	6	1.48	25.52	4.06	3.88	<0.038

RR, relative risk for the cluster compared with the rest of the region; LLR, likelihood ratio. * Mortality rate from diarrhea and infectious gastroenteritis per 100,000 inhabitants.

**Table 4 tropicalmed-07-00312-t004:** Correlation between the mortality rate from diarrhea and infectious gastroenteritis and socioeconomic indicators of human development and social vulnerability in the municipalities of Alagoas, Northeast Brazil between 2000 and 2019.

SDH/Socioeconomic Indicators	Rho	95% CI	*p*-Value
Percentage of children aged 0–5 years who did not attend school	0.4012	0.2186 to 0.5566	<0.001
Mortality up to 1 year of age	0.3449	0.1555 to 0.5098	0.004
Social vulnerability index (SVI)	0.3374	0.1473 to 0.5035	0.001
GINI index	0.2976	0.1045 to 0.2920	0.008
Percentage of people in households with inadequate water supply and sanitation	0.2975	0.1036 to 0.4697	0.002
Percentage of children living in households where none of the residents have completed primary education	0.2683	0.0720 to 0.4445	0.006
Illiteracy rate of the population aged 15 and over	0.2334	0.0349 to 0.4141	0.018
Percentage of people aged 18 or over without complete elementary school and in informal occupation	0.2358	0.0374 to 0.4162	0.317
% of people aged 6 to 14 who do not attend school	0.2100	0.0103 to 0.3935	0.034
Infant mortality rate	0.1117	−0.0903 to −0.3050	0.002
Percentage of admissions for primary care-sensitive conditions	−0.0313	−0.2300 to 0.1698	0.754
Percentage of urban population residing in households connected to the water supply network	−0.1829	−0.3822 to −0.0326	0.002
Unemployment rate for the population aged 18 and over	−0.2264	−0.4080 to −0.0276	0.222
Percentage of 18 to 20 years old with complete secondary	−0.2277	−0.4102 to −0.0302	0.020
Percentage of people covered by supplementary health plans	−0.2433	−0.4228 to −0.0454	0.018
Percentage of people covered by supplementary health plans	−0.2433	−0.4228 to −0.0454	0.013
Percentage of children aged 5 to 6 years in school	−0.2803	−0.4549 to −0.0850	0.004
Per capita income of vulnerable to poverty	−0.3289	−0.2101 to −0.5302	0.005
Municipal human development index (MHDI)	−0.3455	−0.5103 to −0.1562	0.004
Life expectancy at birth	−0.3470	−0.5116 to −0.1579	0.001

SDH = social determinants of health; 95% CI = confidence interval; correlation analysis was performed with the Spearman test (Rho).

## Data Availability

Not applicable.

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
