# Peer review of "Space–Time Clustering and Socioeconomic Factors Associated with Mortality from Diarrhea in Alagoas, Northeastern Brazil: A 20-Year Population-Based Study"

_tropicalmed, 2022, doi:10.3390/tropicalmed7100312_

Round 1

Reviewer 1 Report (Previous Reviewer 1)

The authors have addressed the comments appropriately. The MS can be accepted at the current form.

Author Response

Dear reviewer, thank you very much for your suggestions and contributions.

Reviewer 2 Report (Previous Reviewer 2)

Most of my comments from the first round have been addressed. I have the following comments on this version (some of which have been left unaddressed from before).

I find the use of the words "color", "yellow" and "brown-skinned" to be highly inappropriate ways to describe race and ethnicity. Please coordinate with the journal editors on this, but I suggest replacing "Ethnicity/color" with "Race and ethnicity", and the categories could be "White, Black/Afrodescendant, Asian, Indigenous, Other non-white".

The 20 socio-economic indicators are not listed and defined among the "variables and indicators" in the methods section. If it's not an efficient use of space to list each one separately, then I suggest putting "v. A suite of 20 municipality-level socio-economic indicators taken from the Atlas of Human Development etc. (table 4.)."

"Herein" and "Herewith" are archaic usages.

Panel B in figure 1 is not neccessary. I strongly suggest removing it and keeping only panels A and C (and relettering them, of course).

Author Response

Dear reviewer, thank you very much for your suggestions and contributions.

All changes were duly carried out.

Reviewer 3 Report (Previous Reviewer 3)

No further comments. The authors have addressed the comments raised previously.

Author Response

Dear reviewer, thank you very much for your suggestions and contributions

This manuscript is a resubmission of an earlier submission. The following is a list of the peer review reports and author responses from that submission.

Round 1

Reviewer 1 Report

The manuscript entitled “Space-time clustering and socio-economic factors associated with mortality from diarrhea in Alagoas, Northeastern Brazil: a 20-year population-based study” applied spatial and temporal analysis to evaluate the patterns of mortality caused by acute diarrhea and potential socio-economic risk factors. Though the methods and conclusion are sound, some issues should be clarified.

Major issues:

1.      All the figures are missing. Not sure the mistake is from the authors or journal.

2.      The inference of retrospective and prospective clusters needs to be discussed clearer in the results.

3.      Please double check the mortality rates in 20-39 yrs group between 2000 and 2019. The APC indicated decreasing pattern does not make sense.

4.      Lines 300-304. The discussion for the increasing pattern between 20-39 is not reliable. If you consider HIV infection is the explanation. The higher HIV infection rate in the 30-59 (63.1%) does echoed to the mortality in your analysis (3.44à1.76)  

 Minor issues:

Line 114. It’s confusing that the 2 references for variables?

Author Response

We appreciate very much for the kind consideration of our manuscript for publication. We thank the reviewers for their kind comments. We have tried to reply to the comments of the reviewers to our best as shown underneath and also included in the text.

We hope that our manuscript is now fit for publication.

Thanking you in advance.

Álvaro Francisco Lopes de Sousa

Global Health and Tropical Medicine (GHTM), Instituto de Higiene e Medicina Tropical, Universidade Nova de Lisboa, Portugal

COMMENTS AND SUGGESTIONS FOR AUTHORS:

Reviewer #1:

The manuscript entitled “Space-time clustering and socio-economic factors associated with mortality from diarrhea in Alagoas, Northeastern Brazil: a 20-year population-based study” applied spatial and temporal analysis to evaluate the patterns of mortality caused by acute diarrhea and potential socio-economic risk factors. Though the methods and conclusion are sound, some issues should be clarified.

Major issues:

  1. All the figures are missing. Not sure the mistake is from the authors or journal.

Answer: We apologize that the figures were not available. In fact, we submitted all figures to the journal. We believe there may have been an error sending the file to reviewers.

  1. The inference of retrospective and prospective clusters needs to be discussed clearer in the results.

Answer: We thank the reviewer for this important suggestion. We believe that as we approached this result together, this could generate misunderstanding. To avoid this and also for the better understanding of the readers, we have separated in different paragraphs the results of the retrospective and prospective spatiotemporal analysis. In addition, we emphasize the relevance of prospective results (lines 248 and 249).

  1. Please double check the mortality rates in 20-39 yrs group between 2000 and 2019. The APC indicated decreasing pattern does not make sense.

Answer: We thank the reviewer for this important point. We've checked the APC of this category and it doesn't show as decreasing, but as stable.

  1. Lines 300-304. The discussion for the increasing pattern between 20-39 is not reliable. If you consider HIV infection is the explanation. The higher HIV infection rate in the 30-59 (63.1%) does echoed to the mortality in your analysis (3.44 à 1.76).

Answer: We thank the reviewer for this important point. In fact, we hypothesized this relationship to explain these findings of high mortality among adults aged 20-30 years. However, the authors agreed that our data do not allow us to make this relationship and, therefore, we removed this explanation from the discussion.

Minor issues:

Line 114. It’s confusing that the 2 references for variables?

Answer: We thank the reviewer for this point. We removed the two quotes from this topic.

Reviewer 2 Report

Congratulations to the authors on a well-executed analysis that is fairly clearly explained and addresses a public health concern of considerable relevance in a population experiencing relative disadvantage. My suggestions are mostly to improve clarity.

The formatting of the in-text citations is inconsistent (some numerals are missing square brackets).

Introduction

Well written (despite some translation problems), with the aims of the study clearly stated.

Methods

Outcome definition is clearly stated, as are the spatial units (municipalities), but the temporal units should also be stated here (years). State in the same paragraph that the study used secondary data collected routinely by health information systems.

Swap the first two paragraphs so that it first describes the study area/scope and then the design. Consider separating this section into “Study area and scope” and “Analysis design” or something. I do not find the area of the state to be helpful information (especially not to three decimal places). Relative size would be better (e.g., “Alagoas is the Xth largest state with the Xth largest population”).

I would merge “Data sources and “collection” with “variables and indicators” so that you state the sources of each variable as you list them. Also, clearly state that the outcome/response variable was the diarrhea-attributable mortality rate calculated from two data sources – SIM and IBGE. What were the 20 socio-economic indicators that were assessed for correlation? These should be listed among the variables and indicators.

Panel B in figure 1 is not necessary. Just make Alagaos red in panel A and you will be able to see it perfectly clearly.

Please explain what “inflection points” are. The spatial analysis is explained much more clearly than the temporal.

Results

Are the terms “Color”, “yellow” and “brown-skinned” used in Brazil? They seem inappropriate to an international audience. Please consult with the editors on the journal’s policies regarding this.

Discussion

This is well articulated, situating the findings within the current evidence base. Limitations are adequately addressed. Some review of the English language is needed.

Author Response

Reviewer #2:

Congratulations to the authors on a well-executed analysis that is fairly clearly explained and addresses a public health concern of considerable relevance in a population experiencing relative disadvantage. My suggestions are mostly to improve clarity. The formatting of the in-text citations is inconsistent (some numerals are missing square brackets).

Introduction

  1. Well written (despite some translation problems), with the aims of the study clearly stated.

Answer: We thank the reviewer for this important point. We certify that the manuscript was submitted for review by a native speaker of English.

Methods

  1. Outcome definition is clearly stated, as are the spatial units (municipalities), but the temporal units should also be stated here (years). State in the same paragraph that the study used secondary data collected routinely by health information systems.

Answer: Thanks to the reviewer for this comment. The study period was already included in the method in the topic “Data source and collection”. However, we also added it in the “Design and study area” topic.

  1. Swap the first two paragraphs so that it first describes the study area/scope and then the design. Consider separating this section into “Study area and scope” and “Analysis design” or something. I do not find the area of the state to be helpful information (especially not to three decimal places). Relative size would be better (e.g., “Alagoas is the Xth largest state with the Xth largest population”).

Answer: We thank to the reviewer for this important comment. However, we decided to keep the topic order and name as is, as it is the standard for observational studies using STROBE (Strengthening The Reporting Of Observational Studies In Epidemiology). Furthermore, most observational studies, particularly those of the ecological type, which we used as a basis for the discussion in this article, also used the same pattern.

Regarding the size of the area in the state of Alagoas, we agreed and removed the unit of measurement (despite its use in ecological studies being standard) and used a relative size, but we left the population quantitative, as we consider it essential.

  1. I would merge “Data sources and “collection” with “variables and indicators” so that you state the sources of each variable as you list them. Also, clearly state that the outcome/response variable was the diarrhea-attributable mortality rate calculated from two data sources – SIM and IBGE. What were the 20 socio-economic indicators that were assessed for correlation? These should be listed among the variables and indicators.

Answer: Thanks to the reviewer for this comment. However, we decided to leave the topic name as is, as it is the standard for observational studies using STROBE (Strengthening The Reporting Of Observational Studies In Epidemiology).

Regarding the response variable for SDH, in the topic “Correlation analysis between deaths from diarrhea and gastroenteritis and SDH” it is stated that the mortality rate was used as a dependent variable. In addition, the topic “Variables and indicators” shows how the rate was calculated.

Regarding socioeconomic indicators, the authors decided not to include the list of indicators in the method. Since the complete list appears in table 4 in the results, this information would become redundant and occupy the space of the main text of the article.

  1. Panel B in figure 1 is not necessary. Just make Alagaos red in panel A and you will be able to see it perfectly clearly.

Answer: The study area map was constructed in this way in order to emphasize the Northeast Region of Brazil and make it easy for the reader to identify its location when reading the discussion. This region has important social and economic inequalities that justify the occurrence of several neglected tropical diseases, in this specific case, diarrhea and infectious gastroenteritis.

  1. Please explain what “inflection points” are. The spatial analysis is explained much more clearly than the temporal.

Answer: Thanks to the reviewer for the question. The explanation for “inflection points” was placed in the topic of temporal analysis (lines 128-132). “This method allows verifying changes in the indicator trend over time by adjusting data from the smallest number of possible joinpoints (inflection points) and tests whether the inclusion of more joinpoints is statistically significant. Thus, time series can present an increasing, decreasing, or stable trend and even different trends in sequential sec-tions”.

Results

  1. Are the terms “Color”, “yellow” and “brown-skinned” used in Brazil? They seem inappropriate to an international audience. Please consult with the editors on the journal’s policies regarding this.

Answer: Thanks to the reviewer for this comment. The terms used to designate skin color in the article are standard and official in Brazil. These terms are available for verification on the website of the Brazilian Institute of Geography and Statistics (IBGE) https://www.ibge.gov.br/

Discussion

  1. This is well articulated, situating the findings within the current evidence base. Limitations are adequately addressed. Some review of the English language is needed.

Answer: We thank the reviewer for this suggestion. The text of the article has been carefully revised.

Reviewer 3 Report

The authors have produced a good manuscript demonstrating the use of spatial temporal analytic methods. The authors, however, should consider clarifying the following;

a) Clarity in the description of the data used in the study, was the data case-based? If not, was the data aggregated along the categories noted in the paper. Was there attempts to look at the coverage for reporting of mortality data within the Mortality Information System (SIM) this might have implications if the rates are different across the years.

b)The retrospective and prospective clusters have overlapping periods and some time points are missing. What are the implications of having the different cluster time periods.

c) The spatial temporal maps are not in the manuscript, but authors make reference to  (Figure 2A-C) but not available for review

d) The author's hypothesis that "...that these deaths may be related to the increase in HIV/AIDS cases among young adults in the state of Alagoas in recent decade..." and the conclusion arrived would need more evidence to arrive at that conclusion.

Author Response

We appreciate very much for the kind consideration of our manuscript for publication. We thank the reviewers for their kind comments. We have tried to reply to the comments of the reviewers to our best as shown underneath and also included in the text.

We hope that our manuscript is now fit for publication.

Thanking you in advance.

Álvaro Francisco Lopes de Sousa

Global Health and Tropical Medicine (GHTM), Instituto de Higiene e Medicina Tropical, Universidade Nova de Lisboa, Portugal

Reviewer #3:

The authors have produced a good manuscript demonstrating the use of spatial temporal analytic methods. The authors, however, should consider clarifying the following:

  1. Clarity in the description of the data used in the study, was the data case-based? If not, was the data aggregated along the categories noted in the paper. Was there attempts to look at the coverage for reporting of mortality data within the Mortality Information System (SIM) this might have implications if the rates are different across the years.

Answer: we thank the reviewer for this important question. Study data were obtained from the Mortality Information System (SIM). SIM is the only system in Brazil responsible for collecting and storing information on mortality in the country. It is an official system regulated by the Brazilian Ministry of Health. All public and private health services in the country (hospitals, basic health units, clinics, among others) make all mortality information available to SIM, through data collected by the patient's death certificate. Therefore, the coverage of information must be standardized across the entire national territory. In addition, the data are not aggregated, they are made available individually for each municipality in Brazil.

  1. The retrospective and prospective clusters have overlapping periods and some time points are missing. What are the implications of having the different cluster time periods.

Answer: Thanks to the reviewer for this comment. The nature of this spatiotemporal analysis is precisely to identify within the entire study period and study area which are the locations, and which are the periods with the highest risk of a particular disease or illness. Additionally, one of the conditioning factors of this analysis is the non-occurrence of geographic overlap of the clusters, in this way, different spatiotemporal clusters can have the same period of time, but this would not be considered a temporal overlap, as they are different areas and independent clusters. Thus, there is no implication in having this overlap. In addition, there is no missing period, statistically a specific period and specific area of high risk of death from diarrhea and infectious gastroenteritis in Alagoas was identified.

  1. The spatial temporal maps are not in the manuscript, but authors make reference to (Figure 2A-C) but not available for review.

Answer: We apologize that the figures were not available. In fact, we submitted all figures to the journal. We believe there may have been an error sending the file to reviewers.

  1. The author's hypothesis that "...that these deaths may be related to the increase in HIV/AIDS cases among young adults in the state of Alagoas in recent decade..." and the conclusion arrived would need more evidence to arrive at that conclusion.

Answer: We thank the reviewer for this important point. In fact, we hypothesized this relationship to explain these findings of high mortality among adults aged 20-30 years. However, the authors agreed that our data do not allow us to make this relationship and, therefore, we removed this explanation from the discussion.
